# Dialogic Model of Prevention and Resolution of Conflicts: Evidence of the Success of Cyberbullying Prevention in a Primary School in Catalonia

**DOI:** 10.3390/ijerph16060918

**Published:** 2019-03-14

**Authors:** Beatriz Villarejo-Carballido, Cristina M. Pulido, Lena de Botton, Olga Serradell

**Affiliations:** 1Faculty of Psychology and Education, University of Deusto, 48007 Bilbao, Spain; 2Department of Journalism and Communication Studies, Universitat Autònoma de Barcelona, 08193 Bellaterra, Spain; cristina.pulido@uab.cat; 3Department of Sociology, Universitat de Barcelona, 08034 Barcelona, Spain; lenadebotton@ub.edu; 4Department of Sociology, Universitat Autònoma de Barcelona, 08193 Bellaterra, Spain; olga.serradell@uab.cat

**Keywords:** cyberbullying, intervention, school, minors, families, teachers, successful educational action

## Abstract

This article analyses the evidence obtained from the application of the dialogic model of prevention and resolution of conflicts to eradicate cyberbullying behaviour in a primary school in Catalonia. The Dialogic Prevention Model is one of the successful educational actions identified by INCLUD-ED (FP6 research project). This case study, based on communicative methodology, includes the results obtained from documentary analysis, communicative observations and in-depth interviews. The evidence collected indicates that the implementation of this type of model can help to overcome cyberbullying; children are more confident to reject violence, students support the victims more and the whole community is involved in Zero Tolerance to violence.

## 1. Introduction

Data on the prevalence of cyberbullying among minors indicate an urgent need for scientific evidence regarding how to prevent such bullying as early as possible [1,2,3], particularly in primary and secondary school. There is abundant scientific literature [4,5,6] that describes how cyberbullying affects young people in different countries. However, there is less scientific literature that describes successful prevention programmes. As indicated by Della Cioppa, O’Neil and Craig [7], there is a current need to develop educational interventions aimed at overcoming cyberbullying based on prevention programmes that have been scientifically proven to be successful. To this end, Della Cioppa et al. [7] evaluated 20 prevention programmes, few of which had been previously evaluated and even fewer of which use scientific evidence. Those programmes that achieved scientifically acceptable results coincided with involving the educational community as a whole or incorporating contexts beyond the school [7].

Recently, the systematic and meta-analytical review conducted by Gaffney et al. [8] provided evidence of the effectiveness of some cyberbullying prevention programmes. This study focused on the analysis of 24 publications; most of them used randomised controlled trials to evaluate cyberbullying prevention programmes and the other ones were quasi-experimental designs with before and after measures. Some of the contributions selected for advancing in this field are related to the fact that the evaluated programmes presented evidence of reducing cyberbullying. However, there is a research need to identify the components of intervention programmes which are most effective in reducing cyberbullying and victimisation and to explain the influence of overlapping offline and online victimisation [8].

The objective of this article is to respond to this urgent research need. Based on an analysis of the scientific literature, the elements that a prevention programme must include to guarantee favourable results were identified. Our in-depth study of the scientific literature focused on prevention programmes based on the behaviour of “bystanders”, which in our context is translated as the role of “observers” or “spectators”. In addition, we examined models of community intervention and presented the results of the application of the dialogic model of prevention and resolution of conflicts (DMPRC, hereinafter) [9] in a specific case. This model has been endorsed as a successful educational initiative by the integrated research project INCLUD-ED of the European Commission’s sixth Framework Programme. INCLUD-ED is the only social sciences research project on the European Commission’s list of the ten most successful research projects in Europe [10]. In addition to this, a qualitative case study was conducted using a communicative method [9].

### 1.1. Children’s Media Use, A Balance Approach

Minors today live in a media-inundated society. As noted by Aguaded [11], the most intelligent and rewarding response to such a society requires the development of media competence. The concern of families has increased in recent years due to the use that minors make of media and the risks such use involves. As indicated by Livingstone et al. [12], it has become impossible for parents to prohibit the use of media. Thus, according to studies conducted by Livingstone and colleagues, families and schools should seek to maximise the opportunities to use media for learning and fun. It is necessary to minimise the risks and invest in the education of families with the aim of creating common strategies, face the risks and increase the positive use of media. We must also include the perspective of the rights of minors regarding digital media while avoiding a vision of minimising risk or adopting an alarmist view. To this end, a balanced approach should be sought [13]. Similarly, other researchers have indicated a need to reinforce the idea of teaching children whom to befriend in social networks. For example, this would include choosing those who treat others well and ignoring those who treat other schoolmates with contempt. These efforts should be part of a wider attempt to create a secure online environment for minors [14]. In addition, it is necessary to remember that cyberbullying among equals is not the only risk that minors can face online. For example, minors may find themselves in situations of vulnerability and become trapped in illegal activity through contacts initiated online. Preventing such problems is part of the contemporary struggle to develop an international joint programme to protect minors [15]. Minors should also be educated on how technologies can be used for social change and be familiarised with how such change is being promoted through, for instance, the use of online platforms to protect the rights of young women [16].

### 1.2. Cyberbullying Does not Occur Alone, It Coexists with other Types of Violence

There is extensive literature dealing with cyberbullying among minors. The consequences of such bullying can be severe and in the worst cases, may disturb the happy and harmonious development of girls and boys, as shown by the number of suicides of young people linked to cyberbullying [17]. Therefore, several authors emphasise the importance of regarding cyberbullying as an issue that also affects health [18]. A study on children between the ages of 14 and 17 in six European countries found that 21.4% had suffered episodes of cyberbullying in the last twelve months. According to the same study, such experiences were more frequent among girls than boys (23.9% vs. 18.5%) [18]. Moreover, recent developments point to the need of seeing the whole picture of violence, since cyberbullying coexists with other types of violence (bullying, sexual harassment, etc.) and they need to be treated together [19,20]. In fact, cyberbullying includes different types of violent interactions (flaming, harassing, cyberstalking, spreading rumours, among others) [21]. In our study, we paid attention to all of them. All these types of violence share a same root; a socialisation on attractiveness linked to violence [22]. This type of socialisation occurs where social interactions (media, peers’ group, family, school) generate a type of socialisation that links attractiveness to violence. Therefore, prevention programmes based on this contribution aim at promoting social interactions where attractiveness is not linked to violence. According to this contribution, the key factor for understanding the whole picture of violence is the type of socialisation that children are learning through their daily interactions [22]. 

### 1.3. Community Intervention and Interactions of All Agents Involved in Educational Centres

Several research teams have investigated digital citizenship training to address violent risks. For example, Jones and Mitchell [23] demonstrated how training that included instruction on respectful online behaviour and active online citizenship encouraged individuals to assume an attitude of active observer in harassment situations, i.e., not to remain passive. 

In recent years, investigations emphasising the urgency of developing programmes that involve the wider social environment have emerged. This includes, not leaving child victims of cyberbullying to themselves but involving their entire social community to get to the root of why cyberbullying occurs [24,25]. Cyberbullying exists because of the aforementioned socialisation that links attractiveness to violence [22]. This outcome explains the tendency of cyberbullies to be the most popular members of their group [14]. As a form of social learning, this perception can be changed, and there are schools in which violence has been rejected by involving the entire community [9]. Therefore, strategies to prevent cyberbullying must go beyond the recommendation of parental supervision [26] and focus on the entire community. Prevention programmes should include all the relevant aspects of the problem: From school to families, the intervention of peers, and the use of information and communications technology (ICT) and social networks on the basis of preventive models, not only punitive ones [24,27]. In addition, the non-trivialization of violence and the active stance of the entire educational community are important [28]. This transversal endeavour known as “0 violence from 0 years old” [29] starts from early childhood. The development of empathy is basic to educate individuals into adopting an active and realistic position in the face of cyberbullying, yet it is one of the barriers found in the scientific literature [30]. Individuals must also learn how to act when confronted with cyberbullying [31]. In this sense, it is also necessary to note that cyberbullying prevention should consider the victims and those who defend them. The latter often suffer reprisals for their support, which in the scientific literature is known as second-order sexual harassment [32]. Other authors agree that the minors with greater social capital, that is, “wealth” in their social relations, are the ones who act with a more proactive attitude in response to cyberbullying. Within a group of equals, these minors can encourage a change of attitude in response to such events [33].

When minors know how to play the role of the observer in response to cyberbullying, their effect in curbing cyberbullying increases [34]. Other research describes the impact of involving young people in models of community collaboration to prevent cyberbullying and harassment. These interventions obtain greater results because they reduce the fear of bullying and increase the confidence of minors in the adults who surround them [35]. Other findings indicate the importance of training mentors for young people in the leadership of prevention programmes [36]. The need to belong and participate in the school improves children’s health [37].

The role of adults is fundamental. According to one study, 60% of the victims of cyberbullying seek trustworthy adults when they report abuse [38]. Based on these results, programmes that incorporate an active role for adults in the community reduce episodes of cyberbullying. The action coordinated through a dialogical leadership model develops a more effective transformation [39]. In addition, the involvement of non-academic families is essential [40], especially non-academic women, whose participation is crucial. Their involvement is facilitated by creating dialogical participation spaces [41] as well as the promotion of communicative acts that ensure protective factors [42]. Finally, communication should be highlighted as a basic element. Minors require adult interlocutors who are able to create safe and reliable communication spaces in which minors can communicate the problems they face [43]. The literature review shows positive results on prevention of bullying and cyberbullying through the active role of observers [25,44,45]. Among these programmes, the Green Dot Bystander Intervention Program is highlighted, which was introduced in 26 secondary schools in Kentucky and has been shown to be effective in reducing sexual harassment and other types of violence in the community [45]. Through the *Cyber Friendly Schools* (*CFS*) in Canada, researchers noted that schools and teachers cannot prevent cyberbullying alone. Families, the minors and the community in general must jointly address this goal if cyberbullying is to be prevented both in- and outside the school [44]. Another example is the Medienhelden Programme from Germany, based on the link between cognitive and affective empathy and the prevention of cyberbullying. In this programme, 722 secondary schools with students between 11 and 17 years old were evaluated, finding that an intervention over time reduced the episodes of cyberbullying and increased the feeling of empathy [46]. 

The cyberbullying prevention programmes with the best results involve the entire educational community, encourage zero tolerance for any type of cyberbullying or other type of violent interactions and train minors, teachers and families to collaborate in the creation of spaces for dialogue. It is recommended that educational centres apply only programmes based on scientific evidence and establish a transversal line of action instead of teaching concrete, specific behaviour in the classroom.

Considering these previous contributions, the research questions addressed in this paper are: a) Does DMPRC foster an environment where children feel confident to denounce cases of cyberbullying? Does the coordination of adults (teachers and family) collaborating in a Zero Tolerance to Violence programme create a safer space? And last, does DMPRC reduce violent interactions while increasing active solidarity towards the victims and those who support them?

## 2. Materials and Methods

This research was based on a qualitative case study using the communicative methodology approach [47]. The aim of this methodology is to find those results that improve the living conditions of people, thus guaranteeing the social impact of the research [48]. The main characteristic of this type of methodology is the dialogic construction of knowledge. This means establishing dialogical interactions mediated by an egalitarian dialogue between the research team and the participants [49]. The aim is to contrast the scientific evidence with the life-world knowledge and the experience of the people interviewed, in order to grasp in depth, the social reality studied.

### 2.1. Selection of the Case Study

The selected case study was based on the qualitative analysis of the application of the dialogical model of prevention and resolution of conflicts. [9]. DMPRC, which is characterised by the involvement of the whole educational community (teachers, family, students and other social agents), has two main characteristics. On the one hand, coexistence agreements are reached in an assembly through a dialogic process that involves students, family members and teachers. On the other hand, spaces for dialogue are created with the aim to reject violence and promote interactions that construct a type of socialisation that links attractiveness with nonviolent models [5]. The school creates a coexistence commission that is composed by different community members (students, family, teachers), so they monitor the implementation of the agreement and contribute to the creation of a safer space for all. The model was applied in an educational centre for early and primary school education in Terrassa, a city in Catalonia (Spain). This school has implemented DMPRC since the school year 2014–2015. In this case study, the research focused on analysing the results of the application of this model in cyberbullying detection and prevention. The educational centre, that is a learning community, had been previously studied because of the positive results achieved in a collaboration between teachers and families, including its impact on improving the lives of neighbourhood residents [50]. However, this study provides the first results on cyberbullying prevention obtained in this school. 

### 2.2. Data Collection

The educational centre had been implementing a successful educational programme involving DMPRC for four academic school years. The research team monitored the results of the programme in terms of improving coexistence during ten months in 2017. Among the various outcomes obtained, this article presents those results regarding action taken in response to cyberbullying that occurred in a specific class of the last key stage of primary school. The analysed data were collected using three research techniques: Documentary analysis of the data on the programme, communicative observation of coexistence commission and in-depth interviews. The documentary analysis included, basically, the documents related to the results obtained by the school, and the documentation related to the implementation of DMPRC. The communicative observation of the coexistence commission at the centre (N = 6), was conducted by one of the researchers, who participated in all of them. Five observed commissions were held at the educational centre and were attended by 18 people on average—students, teachers and family members. The sixth communicative observation was held on “Zero Violence” workshop, and 90 people attended. Students that participated in these gatherings explained how the coexistence agreement was being implemented and which were the problems that they had to face in order to guarantee an environment free of violence. Then, an open dialogue between students, teachers and family members was developed, in order to address the problems found. Finally, we developed in-depth interviews (N = 4) with members of the educational community directly involved in the follow-up on the cyberbullying case occurred. The communicative observations facilitated the analysis of the interactions between students, families and teachers at the coexistence commission. In these sessions, the most relevant problems were addressed and joint solutions were found. In one observation session, a case of cyberbullying emerged. This case is discussed in greater detail in this article. In-depth interviews were conducted with several people: The person responsible for promoting coexistence, the teacher of the class in which the cyberbullying case occurred, the centre’s director and the mother of a student. The aim of the interviews was to increase our understanding of the different roles of responsibility adopted regarding the action taken, and the results that were achieved.

### 2.3. Ethical Issues

The case study was developed following the ethical criteria developed in the research “Saleacom. Overcoming Inequalities in Schools and Learning Communities: Innovative Education for a New Century” funded by the European Commission (Reference number: 645668).

## 3. Results

Our research results suggested three main achievements. The first was how the implementation of DMPRC helped break the silence in situations of harassment. In the specific episode of cyberbullying analysed, breaking the silence involved getting children to empower themselves and denounce the cyberbullying cases. The second achievement was the improved intervention by the adult community in response to cyberbullying, which fostered the creation of a safe environment where minors gained sufficient trust to bring up problems that occurred outside the school. The third achievement was active solidarity with the victims and the individuals who support them, thus succeeding at the prevention of re-victimisation. 

### 3.1. From Silence to an Active Stance against Cyberbullying

The coexistence commission was mixed assemblies involving families, teachers and students. They addressed daily problems with respect to coexistence, how to act before such problems and how to prevent them. The coexistence commission can be described as the space for dialogue where the application of DMPRC is monitored. One goal supported by students, family members (who were primarily non-academics) and teachers was achieved; coexistence was improved in all areas since they began to apply DMPRC. Another highly important result was that both girls and boys had developed the confidence to report cyberbullying or other types of violent interactions. According to the researched participants, such reports had not occurred in the past. Therefore, the first research question is confirmed, since the dialogic model of prevention of conflict helped children denounce cyberbullying and feel more confident to do so. An example of this reporting was shared by the teacher of a class group in which cyberbullying was identified.

“One day, a child in my class tells me that he wants to talk to me and that he will come with another classmate. So, they tell me that there has been a situation of cyberbullying through the homework chat tool they have in class, and then, he tells me at three o’clock I’ll bring the mobile and show it to you. (...). And yes, when they showed the conversation, some girls were insulting another saying that nobody could stand her, that she was ugly and stupid, and they kept laughing at her. And he intervened, saying, ‘what you are doing is cyberbullying’, and another answered that it was a lie; and since he saw that they did not stop, he decided to tell me about it.”.(Antonia)

As explained by the teacher in an interview, this student intervened in the WhatsApp group conversation because of the constant dialogue held in the school regarding such interactions and how to act in a courageous manner. According to his teacher, when asked why he was so determined to denounce the situation, the student answered, “because you are brave if you denounce, and if you do not do it, you’re not” (Maria). This is an action that empowers the educational community to act, and that can improve the physical and emotional wellbeing of the victim and the people who support him. His teacher also added, “The victim is fine; the peers treat her well with respect. (...) in the case of the child who reported the cyberbullying case, the class has admired him” (Maria, teacher). According to the interviewee, in this manner, it has been possible to create a climate in the school where silence is not desired when cyberbullying or other types of violent interactions occur. On the contrary, instead of silence, actively facing harassment has become attractive. According to the interviewees, girls and boys who perceive this new climate feel empowered and dare to talk about cyberbullying or other aggressive interactions, something that did not occur in the school before DMPRC was implemented. 

### 3.2. More Secure and Trusted Environments when Adults are Working in the Same Direction with Evidence

Regarding the question of whether or not the coordination of adults (teachers and family) collaborating in Zero Tolerance Violence creates a safer space, the answer is affirmative, in line with the scientific findings in this field. According to the data analysed, students feel secure at school, they treat coexistence problems as a community concern and the fact of seeing their families and teachers together increases their confidence. One of the contributions explained by the coordinator of the coexistence commission was that students also talk about the problems that they have outside school:
“You think that the increase in the security feeling is due to the fact that now they also turn to us with problems that occur with other children in the neighbourhood”. Similarly, the centre’s director highlighted the students’ trust in the educational community: “The students feel that they have the support of the adults who care for them and protect them, an environment they can count on”. In another example, the interviewed mother noted that, “ever since the problem of cyberbullying was discussed in the gathering, teachers, students, family members, children are much better”.(Hamida)

In this sense, the interviewees indicated that working together to eradicate violence creates a good atmosphere achieved in various domains within the school: In the coexistence commission, in the training of family members and teachers, and in the daily dialogues that occur in the school. The positive effect achieved in all these domains has been guided by scientific evidence that supports DMPRC and the validity of dialogic pedagogical gatherings. For the coexistence coordinator, the role played by this type of discussion was key to establishing harmony in the school’s approach to preventing violence and teachers’ actions according to this principle:
“(...) The process of reading is fundamental: Scientific evidence, reading and sharing, and implementing. Without that part of the training, we would not succeed because everyone would go with his or her own preconceived ideas, and we would not be so successful. (...)”.(Montserrat)

Therefore, one key to success was to address the problem using scientific evidence. Another one was to engage the entire community in dialogue to agree on common criteria for action, both in the family and at school.

### 3.3. Active Solidarity toward Victims and the Individuals who Support Them

Last, we highlight an important result regarding a change of attitude toward victims and the individuals who support them. Regarding the question of whether or not DMPRC reduces cyberbullying and increases active solidarity with the victims and those who support them, the answer is affirmative. According to the interviewees, in the past, when one student bullied another, other students that saw this behaviour chose to be quiet because they felt afraid of the consequences of supporting the victim, for instance being bullied too. Some students even decided to engage in the bullying in order to be more popular. Now, since the application of DMPRC, this behaviour has changed. One teacher explained an attack on a child who reported an episode of cyberbullying: “The day after denouncing the cyberbullying and making it public, one of the children was attacked; someone had written *rat* in his book” (Antonia). According to the teacher, in response, the class was stopped and the students discussed the problem. The class group demonstrated solidarity with the child who had denounced the cyberbullying case because the child had shown courage, and the student who wrote the insult was left without support. In this way, students learn to reject violent behaviour by practicing solidarity, and the bullies change more quickly because they do not find social recognition from their peers, from the families or the teachers. No one supported the violent behaviour, according to the interviewees. 

## 4. Discussion

The literature review suggested that research needs to provide more evidence on successful programmes addressed to prevent cyberbullying cases in schools [7,8]. Recent findings suggested the need to focus on the components of intervention programmes which are most effective in reducing cyberbullying perpetration and victimisation [8]. This case study provided evidence that contributes to advances in this research field. The dialogic model of prevention and resolution of conflicts implemented the recommendation of involving the whole community (students, teachers and family) [24,25,45] and obtained excellent results.

This involvement is characterised by a dialogic process where members of the community reach agreements on coexistence and monitor the application of such agreements through the coexistence commission. In this commission, students have a prominent role supported by adults of the community (teachers and family) [9]. By seeing that their reference adults work together for eradicating all types of violence, children feel more confident to report cyberbullying episodes. The school or families did not prohibit the use of media; in fact, in this case students have a WhatsApp group for homework. It was in this group that the cyberbullying episode occurred. Nevertheless, adults do not leave students alone with those online interactions but apply the balance approach of children media use [13,14]. Adults accompany this use through dialogic intervention. For this reason, when one child saw the cyberbullying episode, he felt the courage to explain it to his teacher, because bystander attitudes [25,44,45] are encouraged by the community every day. However, one of the components that none of the programmes analysed in the literature review explained is the second characteristic of DMPRC. This second characteristic focuses on the promotion of interactions that reject violence, while promoting interactions that construct a type of socialisation that links attractiveness with nonviolent models [9]. 

This contribution overcomes one of the barriers found in the literature review; the fact that cyberbullies tend to be considered among the most popular members of their group of peers [14]. If children feel that being a cyberbully makes you more attractive and popular, the prevention programme will always fail. For this reason, one of the success components found in the application of this dialogic model of prevention and resolution of conflicts is that it addresses this problem by including interactions that promote attractiveness towards nonviolent attitudes. Thus, the scenario is reversed, and the brave and popular students are those who reject violence. An example of this can be seen in the fact that one of the students increased his attractiveness after rejecting the violence in the WhatsApp group and denouncing the cyberbullying case. One of the cyberbullies tried to attack him saying that he was a rat, but instead of being bullied by the other peers, the group faced this attack by supporting him and the victim of cyberbullying. Therefore, the solidarity attitude increased between peers, and the child who bullied learnt that he had no more peer support for his behaviour. This way is one of the most effective manners to change this behaviour according to the analysis of the literature review and the fieldwork done. This last contribution helps to advance the research of effectiveness of components on cyberbullying prevention programmes.

The limitation of this research relates to the fact that the in-depth interviews were conducted only with adults. Even though student voices were heard indirectly in the observations and interviews, future research should seek to gather students’ opinions in a more direct manner, for instance through communicative discussion groups. This would allow gaining insight in their results’ contribution of the dialogic model of prevention and resolution of conflicts that faces cyberbullying as well as other types of violent interactions.

## 5. Conclusions

Despite this limitation, the article contributed to the scientific literature by offering qualitative evidence of the impact of applying the dialogical model of prevention and resolution of conflicts. Without the programme, the girl who was cyberbullied in the WhatsApp group would continue suffering this violence today. Yet, now, she feels protected and has more friends willing to show their support. At the same time, the evidence contributed to the crucial role of bystanders, by showing that when their action is seen as brave and attractive, its effectiveness is increased. In addition, all the children in the school feel more confident in their adult environment than they did before. They do not hide what occurs, and they show trust. They know that to denounce it is to be brave and when facing a problem, they seek the support of their teachers and families in order to find a solution all together, as they stated in the interviews. Future research can examine more in depth the type of interactions that successfully achieve the eradication of all types of violence, including cyberbullying interactions.

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
