# Peer review of "Dialogic Model of Prevention and Resolution of Conflicts: Evidence of the Success of Cyberbullying Prevention in a Primary School in Catalonia"

_ijerph, 2019, doi:10.3390/ijerph16060918_

Round 1

Reviewer 1 Report

1. Line 26&27 "There is an abundant scientific literature...." . After such a straight statement, it would be great to provide some references.

2. It is still unclear (from results), what exactly methods or action plans, or algorithms and etc. were established and provided in Educational center for preventing cyberbullying and its consequences? 

3. Discussion part could be more relevant to results. There is a lack of interpretation and should be extended by comparing with other similar researches. 

Author Response

Reviewer 1

Comments addressed

1.   Line 26&27 "There is an abundant scientific literature....".   After such a straight statement, it would be great to provide some   references.

We   have added references as suggested:

Athanasiou K,   Melegkovits E, Andrie EK, Magoulas C, Tzavara CK, Richardson C, Greydanus D,   Tsolia M, Tsitsika AK. Cross-national aspects of cyberbullying victimization   among 14–17-year-old adolescents across seven European countries. BMC public   health. 2018 Dec;18(1):800. 

 Baldry AC, Farrington   DP, Sorrentino A, Blaya C. Cyberbullying and cybervictimization.   InInternational Perspectives on Cyberbullying 2018 (pp. 3-23). Palgrave   Macmillan, Cham.

 Palladino BE,   Menesini E, Nocentini A, Luik P, Naruskov K, Ucanok Z, Dogan A,   Schultze-Krumbholz A, Hess M, Scheithauer H. Perceived severity of   cyberbullying: differences and similarities across four countries. Frontiers   in psychology. 2017 Sep 20;8:1524.

2.   It is still unclear (from results), what exactly methods or action plans, or   algorithms and etc. were established and provided in Educational center for   preventing cyberbullying and its consequences?

The   dialogic model of prevention and resolution of conflicts is explained in more   detail. See 214-225

3. Discussion part could be more relevant to   results. There is a lack of interpretation and should be extended by   comparing with other similar researches.

We   have addressed this comment improving the discussion section. See 370-413

Reviewer 2 Report

Article Review Journal: IJERPH Dialogic Model of Prevention:….

Feedback: The article addresses an important issue about tackling cyberbullying in the school system with a methodology that involves participants at the time of the events.

Areas that need improvement 1. The paper needs to be checked for grammar and flow. 2. Material and Methods section:  The first paragraph is confusing. This needs to be rewritten. 3. Authors need to be clear when they shift from the term cyberbullying to harassment and back. See lines 185, 186, 188 199. Is it that the school looks at harassment in general and the study on cyberbullying? Is it the way the school policy is written? Several sentences explaining why these terms are being used interchangeably needs to be provided.  4. Methods section: For the reader the methods need to be explained further. How were the three techniques actualized? Provide more detail.  5. Discussion and Conclusion section: This section is a conclusion. It is not a discussion section. The discussion section should link the findings (which are very interesting) back to the greater body of literature on cyberbullying.  

Author Response

Reviewer 2

Comments addressed

Feedback:   The article addresses an important issue about tackling cyberbullying in the   school system with a methodology that involves participants at the time of   the events.

Thank   you for your comment.

1.   The paper needs to be checked for grammar and flow.

Done.

2.   Material and Methods section:  The   first paragraph is confusing. This needs to be rewritten.

We   have improved this section. See lines 207-2013

3.   Authors need to be clear when they shift from the term   cyberbullying to harassment and back. See lines 185, 186, 188 199. Is it that   the school looks at harassment in general and the study on cyberbullying? Is   it the way the school policy is written? Several sentences explaining why   these terms are being used interchangeably needs to be provided.

We   have improved the manuscript considering this comment. We have introduced a   clarification in the line 98-106. We have reviewed the coherence of the terms   used and unified.

4.   Methods section: For the reader the methods need to be explained further. How   were the three techniques actualized? Provide more detail.

We   have provided more detail on the techniques used as suggested. See 241-254.

5. Discussion   and Conclusion section: This section is a conclusion. It is not a discussion   section. The discussion section should link the findings (which are very   interesting) back to the greater body of literature on cyberbullying. 

We   have created two sections: discussion and conclusion.  We have improved the discussion one   considering comments received. See 370-413.

Reviewer 3 Report

Thank you for the opportunity to review this manuscript. While the research study focuses on an extremely important topic: preventing cyberbullying among adolescents; there were areas where the manuscript could have been improved considerably before resubmission. As a result, I have recommended I have noted some of these issues below, along with general recommendations that might further strengthen the contribution and quality of this manuscript, either within this journal or elsewhere. Please feel free to incorporate these suggestions on board, as you see fit.

There were minor issues with grammar throughout, and sentence structure in some areas could be improved for clarity (see lines: 18; 26-27; 55-57; 96-98; 119-120)

Ideally the review of literature in the introduction could have been structured more clearly, and drawn upon a more comprehensive range of references to support ideas. In particular, the theoretical frameworks around 'socialisation of attractiveness to violence' along with the 'dialogic model of prevention' required more substantial elaboration, given that these were key elements of the present study

The key research objectives could be justified more strongly and clearly. In particular, lines 125-126 contradict lines 33-34 about the lack of community-focused prevention programs. Suggest rewording this, and more strongly emphasising the specific unique contributions of the present study, given that other similar research exists within this sphere

Likewise, further information is required in the methods section, particularly on key concepts such as 'the communicative method'. For instance, it would be good to distinguish how this is different from the regular interview design, and what the benefits or justifications are for using this specific method. Further description around the specific sample and data analysis procedures is also required here.

A sub-section on the ethical considerations and approval for this project is needed

The findings themselves are very brief and discussed in a somewhat shallow and limited manner. Quotations do not provide strong support for some of the claims being made (for instance, lines 217-219 - what does this mean and how is it supporting the previous claim?). Ideally, this section requires more work in terms of elaborating clearly on the key findings, and distinguishing findings from each other (there is some overlap of content).

Likewise, the discussion itself is very brief and does not link back to the main theoretical models (dialogical model; socialisation of attractiveness to violence) in any meaningful way

Suggest reconsidering 'limitation' around qualitative research (line 286). Qualitative research alone is not a limitation and can provide valuable insight into phenomena that may not always be possible using quantitative methods alone. The authors' view suggests that perhaps alternate methods should have been considered or a stronger justification presented for choosing the current methodological approach

Cyberbullying and (sexual) harassment used interchangeably throughout manuscripts, although they are conceptually (and legally) very different. Suggest rewording for clarity

Conflict of interest statement is missing and particularly important with funded research

All the best with your manuscript and future research endeavours!

Author Response

Reviewer 3

Comments addressed

Thank   you for the opportunity to review this manuscript. While the research study   focuses on an extremely important topic: preventing cyberbullying among   adolescents; there were areas where the manuscript could have been improved   considerably before resubmission. As a result, I have recommended I have   noted some of these issues below, along with general recommendations that   might further strengthen the contribution and quality of this manuscript,   either within this journal or elsewhere. Please feel free to incorporate   these suggestions on board, as you see fit. There   were minor issues with grammar throughout, and sentence structure in some   areas could be improved for clarity (see lines: 18; 26-27; 55-57; 96-98;   119-120)

Thank   you for your comment. 

We   have corrected the sentence structure and grammar issues pointed.

Ideally the review of literature in the introduction   could have been structured more clearly, and drawn upon a more comprehensive   range of references to support ideas. In particular, the theoretical   frameworks around 'socialisation of attractiveness to violence' along with   the 'dialogic model of prevention' required more substantial elaboration,   given that these were key elements of the present study

We have improved the structure of this section and   explaining with more detail the two key concepts suggested.

The   key research objectives could be justified more strongly and clearly. In   particular, lines 125-126 contradict lines 33-34 about the lack of   community-focused prevention programs. Suggest rewording this, and more   strongly emphasising the specific unique contributions of the present study,   given that other similar research exists within this sphere.

We   have corrected this sentence and we have added our three research questions to   clarify our research objectives. See line 200-204.

Likewise,   further information is required in the methods section, particularly on key   concepts such as 'the communicative method'. For instance, it would be good   to distinguish how this is different from the regular interview design, and   what the benefits or justifications are for using this specific method.   Further description around the specific sample and data analysis procedures is   also required here.

We   have improved the methods section considering comments suggested.

A   sub-section on the ethical considerations and approval for this project is   needed

We   have added a sub-section including the ethical criteria followed. See line   264-267

The   findings themselves are very brief and discussed in a somewhat shallow and   limited manner. Quotations do not provide strong support for some of the   claims being made (for instance, lines 217-219 - what does this mean and how   is it supporting the previous claim?). Ideally, this section requires more   work in terms of elaborating clearly on the key findings, and distinguishing   findings from each other (there is some overlap of content).

We   have improved the results section considering the comments suggested.

Likewise, the discussion itself is very brief and   does not link back to the main theoretical models (dialogical model;   socialisation of attractiveness to violence) in any meaningful way

We   have improved the discussion one considering comments received. See 370-413.

Suggest   reconsidering 'limitation' around qualitative research (line 286).   Qualitative research alone is not a limitation and can provide valuable   insight into phenomena that may not always be possible using quantitative   methods alone. The authors' view suggests that perhaps alternate methods   should have been considered or a stronger justification presented for   choosing the current methodological approach

We   have modified this sentence considering this comment. See line 408-413.

Cyberbullying and (sexual) harassment used   interchangeably throughout manuscripts, although they are conceptually (and   legally) very different. Suggest rewording for clarity

We   have improved the manuscript considering this comment. We have introduced a   clarification in the line 98-106. We have reviewed the coherence of the terms   used and unified.

Conflict   of interest statement is missing and particularly important with funded   research. All   the best with your manuscript and future research endeavours!

We   have added this statement. Thank  you.

Round 2

Reviewer 3 Report

Thank you for your response and changes made to the manuscript, which have strengthened the overall contribution of this research. 

Author Response

Dear Reviewer,

First, thank you for your comment, effort and collaboration.  Regarding the comment:

"Thank you for your response and changes made to the manuscript, which have strengthened the overall contribution of this research. There are still some minor issues with grammar and syntax that will need to be addressed before the manuscript is accepted."

We have addressed these some minor issues with grammar and syntax in the new manuscript updated.

Best regards.